# Genome-Wide Integrative Transcriptional Profiling Identifies Age-Associated Signatures in Dogs

**DOI:** 10.3390/genes14061131

**Published:** 2023-05-23

**Authors:** Hyun Seung Kim, Subin Jang, Jaemin Kim

**Affiliations:** 1Division of Applied Life Science (BK21 Four), Gyeongsang National University, Jinju 52828, Republic of Korea; doyemi@naver.com (H.S.K.);; 2Institute of Agriculture and Life Sciences, Gyeongsang National University, Jinju 52828, Republic of Korea

**Keywords:** aging, dog, transcriptome, methylome, epigenetics, differentially expressed genes

## Abstract

Mammals experience similar stages of embryonic development, birth, infancy, youth, adolescence, maturity, and senescence. While embryonic developmental processes have been extensively researched, many molecular mechanisms regulating the different life stages after birth, such as aging, remain unresolved. We investigated the conserved and global molecular transitions in transcriptional remodeling with age in dogs of 15 breeds, which revealed that genes underlying hormone level regulation and developmental programs were differentially regulated during aging. Subsequently, we show that the candidate genes associated with tumorigenesis also exhibit age-dependent DNA methylation patterns, which might have contributed to the tumor state through inhibiting the plasticity of cell differentiation processes during aging, and ultimately suggesting the molecular events that link the processes of aging and cancer. These results highlight that the rate of age-related transcriptional remodeling is influenced not only by the lifespan, but also by the timing of critical physiological milestones.

## 1. Introduction

The dog’s significant role as a model species for comprehending genome evolution, population genetics, and the genes responsible for complex phenotypic characteristics can be attributed to its abundant genetic resources and advancement in genome sequencing [1]. Studies have shown that over 11 years of age, 28% of dogs have one or more diseases [2], and olfactory epithelial cells degenerate from 14 years of age or older [3]. Aging in dogs also leads to cognitive abnormalities and similar aging lesions and treatment methods as humans [4], which illustrates that dogs are an important model for studying aging and other related diseases [5]. The longevity of different dog breeds varies significantly, with larger breeds such as the Great Danes and Bouviers typically having a shorter lifespan of around 6 or 7 years, while smaller breeds such as the Chihuahuas and Toy Poodles can live up to 15 years. Dog breeds also show variation in age-associated diseases. For instance, Bernese Mountain Dogs exhibit a cancer mortality rate over 10 times higher than that of other dog breeds, even after adjusting for age and sex [6]. Further investigations specific to this breed have revealed variations in the *CDKN2A/B* gene region that are associated with histiocytic sarcoma, the most prevalent type of tumor observed in Bernese Mountain Dogs [7]. Dalmatians display an elevated risk of urolithiasis, characterized by the formation of urinary tract stones due to the increased excretion of uric acid [8]. Pedigree analyses of Dalmatians have confirmed the heritability of this trait [9], and linkage mapping has identified the *SLC2A9* gene as the locus responsible for hyperuricemia in Dalmatians [10]. Similar to humans, as dogs age, their susceptibility to various diseases and mortality rates increases exponentially [11]. The observed variations in mortality risk and causes of death among these dog breeds highlight the valuable role of dogs as a robust model for investigating both the genetic and environmental factors that influence the aging process, as well as the underlying molecular mechanisms involved. These valuable insights have the potential to enhance our comprehension of aging, not only in canines but also in all mammals, including humans [12]. 

Biological aging should not be perceived solely as a gradual deterioration of organisms. Rather, organisms possess inherent mechanisms of self-repair that aim to counteract physical decline and maintain homeostasis [13]. However, for reasons that remain poorly elucidated, these self-repair mechanisms gradually diminish over time, rendering aged organisms susceptible to damage, deterioration, and mortality [14]. Aging is a complex process that involves the accumulation of harmful changes in cells and tissues [15]. The elderly are at an increased risk of developing severe illnesses such as cancer, cardiovascular disease, senile dementia, and type-II diabetes due to the gradual deterioration of several physiological processes that occur with aging [16]. In the last decade, a mounting body of literature has indicated that aging is associated with gradual alterations to epigenetic information in both dividing and nondividing cells [17]. These modifications manifest at different levels, including reductions in the bulk levels of the core histones, changes in the patterns of histone posttranslational modifications and DNA methylation, the substitution of canonical histones with histone variants, and modifications in noncoding RNA expression, throughout organismal aging, and replicative senescence [18]. Ultimately, the cumulative effects of these epigenetic changes during aging is a shift in the local accessibility to the genetic material, leading to dysregulated gene expression, reactivation of transposable elements, and genomic instability [19]. With the advent of the Next-Generation Sequencing (NGS) technique, the high-throughput study of DNA methylation and gene expression has become feasible, providing valuable insights into the biology of aging and associated diseases [20]. 

More specifically, although the DNA sequence is subject to the forces of natural selection over many generations, certain aspects of the genome can undergo changes within shorter timeframes, including within an individual’s lifespan. One notable type of change involves chemical modifications of the DNA sequence itself, specifically patterns of DNA methylation, which have been recognized as a characteristic feature of aging [21]. 

The initial investigations into the relationship between DNA methylation changes and aging were conducted on various organs and life stages of humpback salmon [22]. These studies revealed a significant decrease in levels of 5-methylcytosine (5mC) during ontogenesis, highlighting the evolutionary significance of DNA methylation. Subsequently, these findings were extended to mammals [23]. Notably, the highest levels of 5mC were observed in embryos, followed by a gradual decline. In rodents, it was demonstrated that the decrease in 5mC was inversely correlated with their lifespan [24]. Additional support for the association between global DNA hypomethylation and senescence was provided by experiments conducted on fibroblast cultures [25]. Furthermore, it was observed that the DNA methylation inhibitor 5-aza-2′-deoxycytidine substantially shortened the lifespan of cells [26], further strengthening the link between global DNA hypomethylation and the aging process. Through the application of epigenetic clocks, these DNA methylation patterns can accurately predict the chronological age and capture elements of the biological age that have implications for mortality risk beyond what chronological age alone can reveal [27]. In model organisms, these epigenetic clocks have demonstrated responsiveness to interventions aimed at extending lifespan, such as caloric restriction, making them valuable biomarkers for assessing how interventions, genotypes, and environmental exposures influence the rates of aging [28]. For example, Thompson et al. found that the age-related correlation between DNA methylation level and age remains conserved across syntenic regions in the genomes of these canid species, as well as in more distantly related mammalian genomes, including humans [29]. A study by Wang et al. further reported the enrichment of epigenetic alterations on extensively conserved modules of developmental genes, where methylation levels revealed a prevailing correlation with advancing age [30].

The transcriptome refers to the entirety of mRNA molecules produced by an individual cell or a group of cells [31]. It serves as a repository of information regarding cell populations, which is encoded in the genome. Consequently, profiling the transcriptome plays a crucial role in deciphering functional genomic elements [32]. Transcriptomics, or the examination of the transcriptome, aims to achieve three main objectives: (1) categorizing all transcripts, (2) analyzing gene expression variations, and (3) quantifying transcript levels across different conditions [33]. Subsequently, the functional characterization of these genes and the transcripts of interest derived from these transcriptome analyses thereby contribute to validating their biological significance [34]. The analysis of age-related alterations in gene expression has been conducted in various organisms, including yeast, flies, worms, rats, mice, and humans [35,36]. These studies have consistently demonstrated a decline in gene expression is significantly associated with the electron transport chain, protein translation, and growth signaling during aging. Conversely, there is an increase in the expression of genes related to innate immunity, inflammation, and DNA damage [37,38]. Nevertheless, it is important to acknowledge that changes in these pathways account for only a fraction of the numerous genes that exhibit altered expression with age, and most observed changes remain unexplained [39]. In human studies, the transcriptional profiles of peripheral blood samples obtained from large cohorts have been utilized to develop expression signatures that exhibit a high correlation with chronological age, regardless of ethnic backgrounds [35]. These expression signatures can serve as transcriptional clocks, effectively indicating an individual’s age [40]. It is worth noting that the construction of these signatures involved iterative correlations with chronological age, aiming to create molecular clocks, rather than unraveling the mechanisms by which gene expression influences the aging process [41]. Notably, different research groups have generated signatures that exhibit limited overlaps with each other and with epigenetic clocks, which are also derived from correlation analyses [42]. However, the strong correlation between these signatures and the chronological age suggests that transcriptomes contain age-related information that remains elusive to our current understanding [43].

In this work, our primary goal was to identify the molecular signatures of aging using transcriptomic profiles. Such signatures, which we define as distinguishing features of molecular changes with age, may be associated with, and play a biological role in the physiological decline that characterizes aging. Therefore, we collected publicly available RNA-sequencing data and investigated the differentially expressed genes involved in aging from various breeds of dogs to find the global genetic markers that could be applied across different dog breeds. Previous research by Wang et al. compared methylation status as an indicator of aging, but it is unclear whether methylation differences affect gene expression levels, and it may only apply to Labrador retrievers [30]. At present, the impact of age-related alterations in DNA methylation on the transcriptome or functionality of a given tissue or cell remains largely unknown due to the limited available data. Therefore, we also collected the whole-genome bisulfite sequencing data to navigate the genetic markers that could be used in combination with the methylation index genes. We contextualized our findings within the backdrop of established age-related physiological and biochemical alterations. Our results additionally expose an array of previously unidentified transcriptional changes in genes, processes, and functions that may serve as prospective targets for future investigations and provide enhanced insights into the relationship between transcriptional changes and the aging process at varying levels.

## 2. Materials and Methods

### 2.1. RNA-Sequencing Data and Analysis of Differentially Expressed Genes

RNA-sequencing data from 15 healthy dogs (15 different purebreds) and 25 healthy mice (C57BL/6J strain) with age information provided were sourced from the NCBI (National Center for Biotechnology Information) [44] and are presented in the Appendix A. For both studies, RNA was extracted from whole blood following the manufacturer’s instructions. For dogs, the age of each sample was standardized in accordance with the life expectancy of the breed detailed by the American Kennel Club (AKC). Trimmomatic (version 0.39) [45] was employed to eliminate low-quality reads. We used the STAR alignment tool to map the reads to the Canfam3.1 (dog) and GRCm38.p6 (mouse) reference genomes, respectively [46] and the read count per gene was estimated using featureCounts [47]. RSEM (ver. 1.3.1) [48] software was used to estimate gene expression levels (counts per million, CPM). The gene expression profile data were then analyzed using the “limma-voom” and “affy” packages of R software (version 4.1.0) [49,50]. By treating normalized age as a continuous variable, differential expression levels were then quantified to identify genes with *p*-value < 0.05, |log2FC| > 1, which were considered statistically significant. Genes with maximal TPM values less than 1 were subsequently excluded. 

### 2.2. Age-Dependent DNA Methylation

Whole-genome bisulfite sequencing data from 104 Labrador retrievers’ whole blood samples were retrieved to estimate the methylation values for all the CpG sites [30]. The sequencing reads were demultiplexed and subjected to quality assessment using FastQC [51]. Next, TrimGalore [52] was employed to remove the first 4 bp of the reads, and the trimmed reads were then aligned to a bisulfite-converted reference genome using Bismark (version 0.14.3) [53] under the parameters “-score min L, 0, −0.2”. The methylation status of the CpG sites was determined using MethylDackel (version 0. 2. 1) (Ryan, 2017) [54]. 

### 2.3. Gene Set Enrichment Analysis

The FUMA GWAS program [55] was used to identify a significant over-representation of genes with particular functional categories, such as the biological process of gene ontology [56] The adjusted *p*-value of 0.05 was used as the criterion for statistical significance.

### 2.4. Figure Visualization

The Manhattan plot was created using the ‘qqman’ and the ‘ggplot2’ package of the R program (version 4.2.2).

## 3. Results and Discussion

### 3.1. Differentially Expressed Genes and Enriched Pathways Associated with Aging

To investigate whether certain genomic regions have a differential impact on changes in transcript profiles during aging, and to examine whether there is a global dysregulation of gene expression, we leveraged the publicly available RNA-sequencing data of 15 dog samples with their age information. These dogs represented 15 different breeds (Rat Terrier, Rottweiler, Miniature Pinscher, Dachshund, Shih Tzu, Collie, Dachshund, Cocker Spaniel, Staffordshire terrier, Australian Shepherd, Standard Poodle, Pekingese, Boxer, Shih Tzu, and Labrador Retriever, respectively), ranging from small to big dog breeds. As the common and surrogate measures of chronological ages from different dog breeds, we thereby weighted the age of each dog with the lifespan for the corresponding breed (Appendix A), using data from the American Kennel Club. We aimed to find a significant correlation between the gene expression level and the measure of aging. 

As a result, a total of 154 genes (77 downregulated and 77 upregulated DEGs, respectively) were found to have changed expression levels significantly with age in dogs (Figure 1 and Appendix A). Analysis of gene ontology (GO) enrichment for DEGs indicated a significant overrepresentation of the categories associated with “hormone transport” (*SLCO1C1*, *EXOC3L1*, *NOS2*, *FKBP1B*, *PRKCE*, *SPP1*, *TNF*, *SLC44A4*, and *SLC16A2*, respectively) and “regulation of hormone levels” (*SLCO1C1*, *EXOC3L1*, *NOS2*, *FKBP1B*, *PRKCE*, *SPP1*, *LRAT*, *TNF*, *SLC44A4*, *PCSK5*, and *SLC16A2*, respectively) (Table 1). Various factors that impact the rate of aging in mammals have been identified [57]. These include well-known endocrine signals such as insulin, growth hormone (GH), insulin-like growth factor-I (IGF-I), and thyroid hormones (THs). Each of these signals play a crucial role in regulating metabolism and influences an organism’s ability to adapt to different environmental conditions [58]. In the process of aging, there is a notable alteration in the secretion patterns of hormones generated by the hypothalamic–pituitary axis [59]. Additionally, the axis becomes less responsive to negative feedback mediated by end hormones. Numerous population studies have reported that serum thyroid-stimulating hormone (TSH) levels increase with normal aging [60,61]. These pathways together may explain the aging mechanisms, particularly the hormonal response to aging, which is governed by altered gene expression and regulation. 

Initial observations have demonstrated a specific link between reduced thyroid function and an extended lifespan in both small and large mammals, including humans [62,63]. As a result, higher levels of serum thyroid-stimulating hormone (TSH) and/or low levels of serum free thyroxine (T4) have been associated with an increased life expectancy, indicating a significant role of thyroid hormones (THs) in the aging process [64,65]. Older rats with similar thyroid-stimulating hormone (TSH) levels as younger rats were found to have a lower DIO1 activity, and research in rodents has indicated that levels of the thyroid hormone (TH) transporter MCT8 in the liver are decreased in aging individuals, suggesting a change in TH responsiveness with age [66]. Additionally, there is evidence suggesting that THs may have a decreased ability to activate certain post-receptor mechanisms involved in thyroid function in older individuals [67]. Therefore, it is noteworthy to mention that genes associated with thyroid hormones *(SLCO1C1*, *SLC7A5*, and *SLC16A2)* showed differential expression patterns (as shown in Appendix A) in the experiment. For example, the TH-activating deiodinase (DIO2) and the transmembrane transporter of thyroxine (*SLCO1C1*) are the primary thyroid hormone transporters at the blood-brain barrier. Reduced thyroid hormone transport across the blood-brain barrier contributes to neurological deficits [68].

It has been observed that epigenetic modifications during the aging process target extensively preserved developmental gene pathways, wherein methylation levels typically increase with age [69,70]. Our findings also revealed that age-dependent genes were significantly overrepresented in terms such as “embryo development” (*GPR161*, *ETNK2*, *ITGA8*, *PDZD7*, *CDON*, *PRICKLE1*, *HOXC4*, *HOXC4*, *PAF1*, *APOB*, *ADA*, *PDGFC*, *FBN2*, *TNF*, *SLC44A4*, and *PCSK5*, respectively) and “embryonic organ development” (*ITGA8*, *PDZD7*, *HOXC4*, *HOXC4*, *ADA*, *PDGFC*, *FBN2*, *TNF*, *SLC44A4*, and *PCSK5*, respectively) (Table 1). Although the biology of aging has traditionally been regarded as distinct from that of development [71], the compelling correlation presented in this study suggests that they are intricately linked. This finding supports the notion that certain elements of the aging process may represent a continuation of these developmental processes, rather than as a discrete phenomenon. 

In dogs, there is a wide range of cancer incidence rates, with the group aged 9–11 years or older exhibiting rates 14 to 32 times greater than those aged 3 years or younger [72]. Most epigenetic changes that occur during the aging process are believed to be programmed and restricted to a subset of tissue cells, potentially making them susceptible to tumorigenesis [73]. Methylation is thought to hinder the plasticity of cell differentiation mechanisms, and likely contributes to the development of tumors [74]. We note that the most significant DEG from our analysis was found in *PARD6* (*p*-value = 2.62 × 10^−4^), as the expression level was found to have increased with age (Figure 2). It is typically found in many cancer tissues, including breast and epithelial cancer, and is considered to play a role in preventing the development of cancer cells by inhibiting cell proliferation and restricting the epithelial cell cycle [75,76]. 

### 3.2. Conserved Signatures across Mammals

To investigate the conserved age-dependent transcripts across these species, we leveraged the publicly available RNA-sequencing data from mice and found 346 significant DEGs (146 downregulated and 200 upregulated genes, respectively) (Appendix A), where we found a modest overlap of five genes (*ITGA8*, *KCNJ1*, *NCR1*, *SLC44A4*, and *PRICKLE1*) between two species. We noted that the common differentially expressed genes have been elucidated for their roles in tumorigenesis. The downregulation of *ITGA8* has been linked to unfavorable overall survival (OS) outcomes in individuals with clear cell renal cell carcinoma [77]. Conversely, in patients with an early relapse of multiple myeloma (MM), high levels of *ITGA8* expression can prompt epithelial–mesenchymal transition (EMT), resulting in increased migratory and invasive properties of the MM cells [78]. The *KCNJ1* gene plays an essential role in potassium balance, and its mutations are associated with several diseases such as prenatal Bartter syndrome and diabetes. The expression of *KCNJ1* suppresses tumor cell growth and promotes cancer cell apoptosis [79]. Studies have shown that patients with prostate cancer exhibit decreased levels of *NCR1*, and that prostate cancer cells themselves suppress the expression of *NCR1*, thus impeding their ability to identify tumor cells. Hypoxic conditions lead to a reduction in the expression of NKp46, ultimately compromising the ability of natural killer cells to effectively eliminate tumor cells [80]. When the expression decreases, the toxicity of natural killer cells against prostate cancer increases, and when the expression increases, it plays the opposite role [81]. In this study, we found the consistent directionality of expression changes between two species during aging in *ITGA8*, *SLC44A4*, and *KCNJ1*, while *PRICKLE1* and *NCR1* showed the opposite patterns. It is not necessarily the case that different species follow the same trajectory of changes as they age. Specifically, dogs exhibit rapid methylome remodeling during early life when compared to humans. Further investigations into this matter suggests that the rate of remodeling is not solely influenced by the lifespan of a species, but also by the timing of the crucial physiological milestones [70]. 

### 3.3. Differentially Expressed Genes with Age-Related DNA Methylation Patterns

Wang et al. characterized the methylomes of 104 Labrador retrievers to report the epigenetic modifications that occurred during the aging process [30]. Regarding gene promoters, hypomethylated CpGs are typically correlated with actively expressed genes that are constitutively expressed, while hypermethylated CpGs tend to be associated with genes that are either silenced or have low levels of expression [82]. We thus integrated the DNA methylation profiles with the transcriptome data to identify the genes that may have altered expression due to DNA methylation, those changes in gene expression that are linked to age-related changes in DNA methylation, and ultimately elucidate the epigenetic regulation of gene expression across a wider range of breeds. Only four genes (*SLC44A4*, *PLCB4*, *HSPB1*, and *PREX2*, respectively) showed concordant results between two independent analyses, and we noted that *SLC44A4* also revealed conserved gene expression changes during aging in mice analysis. *SLC44A4* has been demonstrated to cause deafness [83], and is significantly upregulated in various epithelial tumors, particularly in prostate and pancreatic cancers, resulting in antitumor responses [84]. Methyl groups bind to CpG islands and consequently contribute to the downregulation of gene expression [85]. Older dogs were found to encompass significantly hypomethylated CpG sites within *SLC44A4*, which may have been attributed to the increased gene expression level and a higher risk of developing cancer in aged dogs (Figure 2 and Figure 3). Age-associated DNA methylation changes in canids implicated similar gene ontology categories as those that are observed in humans, suggesting an evolutionarily conserved mechanism underlying age-related DNA methylation in mammals. 

Overall, however, we observed a limited correlation between the age-dependent DNA methylation patterns of Labrador retrievers and gene expression levels across the 15 breeds assessed, which may be possibly attributed to the distinct dog breeds exhibiting widely heterogeneous molecular mechanisms of aging and varying lifespans. The application of recent genomic methodologies has unveiled a more intricate relationship between methylation and gene expression [86]. It is now evident that genes linked to unmethylated CpG islands (CGIs) do not always exhibit expression, while genes associated with methylated islands can indeed be expressed. To further elucidate this complexity, future genomic investigations can be directed toward exploring the involvement of chromatin structures, as well as activators and repressors of gene transcription [87,88]. These experimental endeavors may be required for unraveling the underlying intricacies in the association between methylation and gene expression.

### 3.4. Limitations

Initially, our study employed RNA extracted from whole blood samples, a widely utilized source in which age-related alterations in leukocyte populations have been well-documented. Notably, prior investigations have established that certain T and B lymphocytes exhibit a decline in numbers as individuals age, which may have impacted our results. In a recent investigation conducted by Day et al., age-associated alterations in CpG DNA methylation discovered in various human tissues showed that hypomethylated CpG sites exhibited tissue-specific age-related changes [89]. Another study focusing on blood DNA also indicated a lack of predictive capacity for other tissues [86], emphasizing the necessity of studying more tissues to understand global methylation status throughout the lifespan. Perhaps more critically, it should be acknowledged that the sample size employed in this study was relatively modest, consisting of 15 samples. Undoubtedly, the construction of a more precise and reliable epigenetic and transcriptomic aging clock could be facilitated by utilizing larger sample sizes.

The emergence of “-omics” technologies, which facilitate investigations into gene expression, chromatin modifications, metabolomes, and proteomes, offers a promising avenue to comprehensively characterize the molecular patterns associated with aging. In order to gain deeper insights into the contrasting factors that either promote or counteract the aging process, numerous studies have examined gene expression alterations across various age groups. Hence, rather than solely examining the biological function of genes exhibiting expression drift, it is perhaps more important to investigate their association with chromatin modifications or chromosomal location. Such analyzes can provide valuable insights into the underlying mechanisms that lead to the disruption of gene regulation. Consequently, future research endeavors should focus on developing novel analytical strategies for transcriptomes that consider the distinct biology of aging. These strategies should incorporate co-expression values, chromosomal location information, and histone modification data to distinguish between gene expression changes that are appropriately regulated and those that are dysregulated [39].

## 4. Conclusions

Our study conducted a comprehensive meta-analysis of aging systematically with the aim of identifying molecular mechanisms of aging. From analyzing gene expression data from various dog breeds, we have identified genes and cellular processes that are globally affected by aging at the transcriptional level. Our research has uncovered previously unknown transcriptional changes that occurred during the aging process. These genes and cellular processes can be targeted for future studies to develop aging biomarkers, and to better understand the transcriptional regulation that occurs during aging. Our results suggest that these molecular patterns represent degenerative processes and reflect the transcriptional responses of healthy cells to degeneration. However, the field of epigenomic studies focusing on aging in dogs, as well as other species, is still in its early stages, and further investigations involving larger cohorts of dogs are needed. By gathering more data from these expanded cohorts, researchers can potentially identify novel molecular mechanisms of aging that could be subjected to modifications or interventions. As our knowledge of the aging transcriptome continues to grow, integrating and analyzing data using meta-profiling and integrative approaches will become increasingly valuable for understanding the aging process.

## Figures and Tables

**Figure 1 genes-14-01131-f001:**
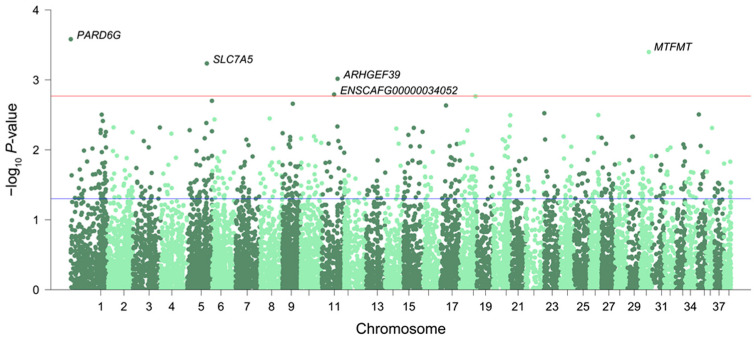
Manhattan plot for the transcriptome-wide significance of expression changes during aging. The threshold for transcriptome-wide significance was *p* < 0.05 and |log2FC| > 1 (blue line), but only the five most significant genes (above the red line) are labeled in this figure. A full list of significant genes is provided in Appendix A.

**Figure 2 genes-14-01131-f002:**
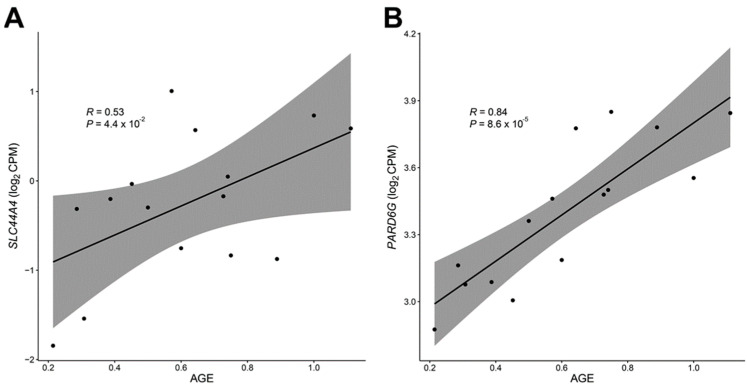
Candidate genes exhibiting transcriptional remodeling with age in 15 dogs. The gene expression level was found to be significantly associated with normalized age measures in the *SLC44A4* (**A**) and *PARD6G* (**B**) genes, respectively.

**Figure 3 genes-14-01131-f003:**
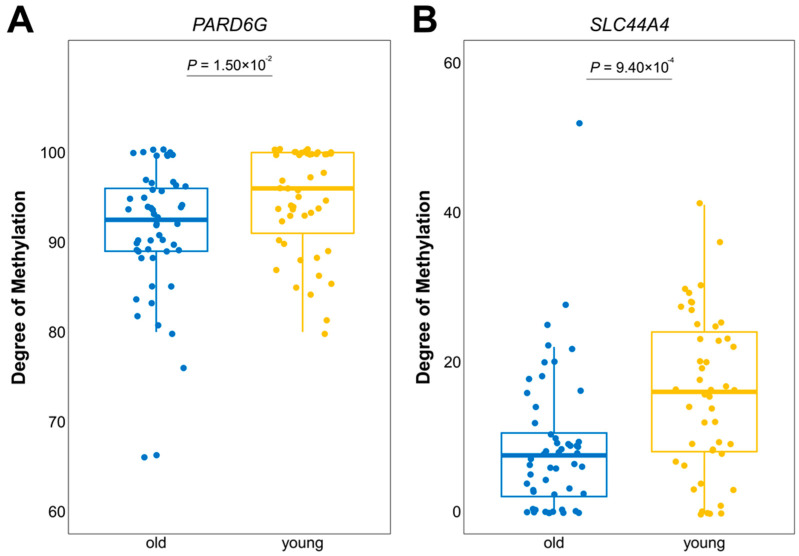
Aging-associated DNA methylation changes in the *PARD6G* (**A**) and *SLC44A4* (**B**) genes. For both candidate genes, hypomethylation was observed in aged Labrador retrievers (6 > years old) compared to younger ones (6 < years old).

**Table 1 genes-14-01131-t001:** Significant enrichment of 154 differentially expressed genes during aging. The table presents pathways that were significantly over-represented (adjusted *p*-value < 0.05).

Pathways	Term	N Gene	N Overlap	*p*-Value	Adj.P
GO-Biological Pathways	Hormone transport	319	9	4.30 × 10^−6^	9.75 × 10^−3^
	Regulation of hormone levels	513	11	5.07 × 10^−6^	9.75 × 10^−3^
	Ion transport	1663	20	5.87 × 10^−6^	9.75 × 10^−3^
	Embryo development	986	15	6.35 × 10^−6^	9.75 × 10^−3^
	Regulation of protein modification process	1826	21	6.63 × 10^−6^	9.75 × 10^−3^
	Regulation of blood circulation	293	8	1.88 × 10^−5^	2.06 × 10^−2^
	Cardiac conduction	143	6	1.99 × 10^−5^	2.06 × 10^−2^
	Regulation of intracellular signal transduction	1824	20	2.24 × 10^−5^	2.06 × 10^−2^
	Ion transmembrane transport	1125	15	2.96 × 10^−5^	2.42 × 10^−2^
	Transmembrane transport	1574	18	3.48 × 10^−5^	2.45 × 10^−2^
	Maintenance of location	325	8	3.93 × 10^−5^	2.45 × 10^−2^
	Embryonic organ development	423	9	4.01 × 10^−5^	2.45 × 10^−2^
	Circulatory system process	541	10	4.86 × 10^−5^	2.75 × 10^−2^
	Regulation of myoblast differentiation	53	4	5.27 × 10^−5^	2.77 × 10^−2^
	Regulation of transmembrane transport	553	10	5.84 × 10^−5^	2.86 × 10^−2^
	Regulation of transport	1827	19	7.43 × 10^−5^	3.40 × 10^−2^
	Regulation of phosphorus metabolic process	1677	18	7.86 × 10^−5^	3.40 × 10^−2^
	Regulation of ion transmembrane transport	467	9	8.54 × 10^−5^	3.49 × 10^−2^
	Heterotypic cell–cell adhesion	61	4	9.17 × 10^−5^	3.55 × 10^−2^
	Multicellular organismal homeostasis	477	9	1.00 × 10^−4^	3.68 × 10^−2^
	Negative regulation of protein modification process	599	10	1.13 × 10^−4^	3.94 × 10^−2^
	Heart process	287	7	1.28 × 10^−4^	4.23 × 10^−2^
	Multicellular organismal signaling	201	6	1.32 × 10^−4^	4.23 × 10^−2^

## Data Availability

Not applicable.

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
