# Peer review of "Genome-Wide Integrative Transcriptional Profiling Identifies Age-Associated Signatures in Dogs"

_genes, 2023, doi:10.3390/genes14061131_

Round 1

Reviewer 1 Report

The study of genetic mechanisms involved in aging -related processes is relevant to develop genomic programs for retard the aging and possibly for increasing lifespan.

The reviewed paper has some obvious limitations. The sample sizes are low. The significance level of gene enrichment is not high as well. However, these limitations cannot be overcome by the revisions within the limited time. However, the paper is interesting, well-written and provides some original results. That is why I assume that the paper might be considered for publication and will be of some interest for potential readers.

The authors should state the limitations of this study in the Discussion and clearly indicate on that the results should be treated carefully and required to be validated on a larger sample.

Reviewer 2 Report

Kim et al. re-analyzed existing RNA-seq and methyl-seq data from dogs and mice to perform analysis on age-related changes in the transcriptome of blood cells. The re-utilization of existing raw data might potentially represent a cost-effective approach to gain new insights into biological processes.

Unfortunately, I consider the starting data in this study insufficient to gain major new insights. The authors attempted to validate their conclusions by comparing data from mice and dogs. However, only 3 genes showed overlapping age-dependent changes in their transcriptomic signatures between dogs and mice. In my opinion, the only plausible conclusion from this result is to state that the analysis failed to provide major new insights into the ageing process. I would not mind to see the publication of such "a negative result", if this is then used to disucss the limitations of the study and potential requirements for more successful future studies. This would be far more useful than the current very speculative statements about the potential function role of the detected candidate genes. There are thousands of papers, in which authors spin plausible stories about the functional role of candidate genes derived from various "omics" analysis. In my opinion, such speculations have very little value without experimental validation and I recommend to delete these parts.

The main focus of the study is on transcriptional changes. As a side aspect, the authors also tried to utilize existing methyl-seq data from Labrador Retrievers (different dogs than those used for the transcriptomic analyses) to correlate epigenetic changes with the transcriptomic changes. While this is also an interesting idea, this part of the manuscript would need a much more thorough introduction. There is already a body of literature on molecular ageing in dogs and wolves and this needs to be properly introduced and cited, if the epigenetic part should be kept.

Specific comments:

Major

(1)

I do not agree with the conclusion that this was a successful study. The limitations of the study should be transparently stated and discussed.

(2)

In its current state, I fail to see in which way the analysis of the methyl-seq data goes beyond what has already been reported in the original Wang et al. paper. This part should be either deleted or it must be sufficiently developed to really yield novel information that has not already been reported in Wang et al.

Other

(3)

Line 25: species --> breed; neutering --> neutering status

(4)

Line 67: "RNA sequencing data extracted from the whole blood ..." Perhaps, the starting RNA was extracted from whole blood, but it is impossible to extract the RNA-seq data from blood. The wet-lab methodology that was used to produce the data must be correctly re-capitualted (very brief, with references to the literature, in which the generation of these data was originally reported).

(5)

Line 82: From which cells or which tissue was the genomic DNA for the methyl-seq experiment derived?

(6)

Figure 1: What is the meaning of the red horizontal line in the figure? The authors state that they used a significance threhold of p=0.05. This is not where the red line is!
